# Detecting Emotions behind the Screen

**Najla Alkaabi** [1,*] , **Nazar Zaki** [1] , **Heba Ismail** [2] and **Manzoor Khan** [1]

1    College of Information Technology, United Arab Emirates University,
     Abu Dhabi 15551, United Arab Emirates
2    Department of CS-IT, College of Engineering, Abu Dhabi University (ADU),
     Abu Dhabi 60511, United Arab Emirates
*    Correspondence: najla.alkaabi.ae@gmail.com or 960223676@uaeu.ac.ae

**Abstract:** Students' emotional health is a major contributor to educational success. Hence, to support students' success in online learning platforms, we contribute with the development of an analysis of the emotional orientations and triggers in their text messages. Such analysis could be automated and used for early detection of the emotional status of students. In our approach, we relied on transfer learning to train the model, using the pre-trained Bidirectional Encoder Representations from Transformers model (BERT). The model classified messages as positive, negative, or neutral. The transfer learning model was then used to classify a larger unlabeled dataset and fine-grained emotions in the negative messages only, using NRC lexicon. In our analysis to the results, we focused in discovering the dominant negative emotions expressed and the most common words students used to express them. We believe this can be an important clue or first line of detection that may assist mental health practitioners to develop targeted programs for students, especially with the massive shift to online education due to the COVID-19 pandemic. We compared our model to a state-of-the-art ML-based model and found our model outperformed the other by achieving a 91% accuracy compared to an 86%. To the best of our knowledge, this is the first study to focus on a mental health analysis of students in online educational platforms other than massive open online courses (MOOCs).

**Keywords:** BERT; education; emotions detection; mental health; NRC lexicon; online learning; sentiment analysis; transfer learning

## 1. Introduction

The business domain has been widely implementing sentiment or emotion analysis to mine users' reactions and to evaluate satisfaction with products or services. New domains such as health, education, and politics are starting to utilize sentiment analysis. Sentiment-analysis studies in education, typically, focus on MOOCs and online-course platforms. These studies aimed to detect sentiments regarding course evaluation [1], system evaluation [2], and the whole learning process [3] or to detect specific emotions related to a student's engagement level [4].

The objective of this study is to determine whether a transfer learning approach to the analysis of students' activity in online educational platforms, other than MOOCS, performs better than a ML approach. To the best of our knowledge, this is the first study to focus on emotions analysis of students in online educational platforms other than MOOCs.

The methodology involves the application of sentiment classification to evaluate messages posted by users of an Android educational app, targeting high-school and college students. The app is open for students from around the world and is not limited to a specific country or region. The app provides high school and university students with access to openly licensed material in various subjects such as physics, chemistry, and calculus. The platform offers users a chatroom to exchange feedback and ask questions.

Over time, we observed that a community of students has developed, and we observed that a few students using the chatroom expressed stress or anxiety and reached out for help. This community had developed due to the fact that students were using the app almost every day or a couple of times per week for a whole semester or an academic year. We believe that this regular usage has helped in building the community and has encouraged students to express their feelings more often. Such dynamic interaction is not usually the case in MOOCs platforms.

In this work, we contribute to the field by testing a two-phase approach for emotions analysis. The approach combines transfer learning using Bidirectional Encoder Representations from Transformers (BERT) [5] and a lexicon-based annotation using NRC lexicon [6] to fine-grain the negative messages into four negative emotions: anger, disgust, fear, and sadness. The main contributions of this work can be summarized as follows:

1.  The study proposes a novel approach that combines transfer learning and NRC lexicon to discover fine-grained emotions.
2.  The study focuses on supporting mental health analysis of university-level students in online educational platforms other than MOOCs.
3.  The study identifies the dominant-negative emotions among a university-level segment of students and the most-used words expressing these emotions. This insight can greatly contribute to designing appropriate support interventions in online education applications.

This section describes the structure of this paper. Section 2 reviews the literature on recent sentiment analysis methods that have been used specifically in transfer learning. Section 3 explains the methodology and approach enacted. It details the data used, the baseline model built, the BERT-based model built, and the NRC-lexicon role in discovering fine-grained emotions. Section 4 showcases the results, specifically the performance test from both models and the analysis of emotions discovery. Section 5 is the conclusion of the paper and presents an outline of the future direction of our work.

## 2. Related Works

Sentiment analysis is an evolving field in text mining research. Applications of sentiment analysis vary and are designed to detect sentiments regarding government decisions and policies, customers' reviews about products or hotels [7], political views [8], health monitoring [9], or financial news [10]. Targeted data sources that are mainly used in this research come from social media platforms (prominently Twitter [1,11]), IMBD movie reviews [12], or travel and hotel platforms [13]. In the past few year, we have started to see more studies on data produced in MOOCs or online courses [1,4]. For many years, there have been two major sentiment analysis approaches in research: lexicon-based and machine learning-based [14]. Lexicon-based methods are expensive and require the intervention of experts. On the other hand, machine learning-based methods are suitable for big data and are less expensive [15]. Hybrid approaches that utilize both methods have also been used. In [16], a genetic algorithm along with a lexicon (SentiWordNet) are used together to achieve better feature reduction for enhancing system scalability. When using the conventional ML algorithms such as Naive Bayes, decision tree, support vector machines (SVM), and logistic regression, the accuracy relies heavily on feature engineering. The widely adopted features in research, when using ML-based sentiment analysis, are bag-of-words, bag-of-characters, character n-grams, words n-grams, part-of-speech (POS), and term frequency–inverse document frequency (TF–IDF) [17].

With advancements in deep learning, we started to notice the shift toward deep learning methods, which can solve complex problems and discover hidden patterns and deep understanding in data. Moreover, deep learning approaches do not rely on feature engineering, which makes it easier to develop and maintain compared to ML. In published research studies, we find models utilizing neural networks, such convolutional neural network (CNN), recurrent neural network (RNN), Long-short-term-memory (LSTM) model, and gated recurrent unit (GRU) [18–20]. To compensate for the shortage of "labeled" data, a recent trend developed and is growing across all tasks of artificial learning, which is

transfer learning. Transfer learning involves the utilization of a pretrained model as part of building a new model. In the field of natural language processing (NLP), BERT is the leading pre-trained model that researchers are currently studying. As the name suggests, BERT is built using a transformer and is designed to pre-train deep bidirectional representations from text by considering both left and right contexts in all layers. BERT-based models use much less data to outperform previous state-of-the-art methods [21].

To transfer knowledge from the pre-trained BERT to downstream tasks, there are two common approaches: (1) the feature-based approach, and (2) fine-tuning. The feature-based approach involves extracting the sentence embeddings from the last hidden layers of BERT and feeding them into another model. On the other hand, fine-tuning is about adding an additional layer on top of the BERT model to finalize the classification. Papers, adopted the fine-tuning approach, used a standard SoftMax layer added on top of BERT to predict the probability of a label. For example, we find in [22] that researchers fine-tuned BERT by adding a softmax layer to classify around 1 million posts as positive, neutral, or negative. Then they used TF-IDF to summarize the topics of negative posts. Another work [23] used BERT to classify 11,855 one-sentence movie reviews extracted from Rotten Tomatoes® as belonging to one of five classes: very negative, negative, neutral, positive, and very positive. For more fine-grained emotions classification, we find [24] used BERT to classify two datasets as neutral, joy, surprise, and anger with accuracy of 86.2% for one of the datasets. They also used a softmax layer.

In the education domain, our area of interest, we find [1] used BERT to classify around 19,000 MOOCS comments, using a shallow BERT-CNN model, as either positive or negative. The results are impressive, with accuracy, F1(Positive), and F1(Negative) of 0.925, 0.950, and 0.801, respectively. On the other hand, researchers who adopted the features extraction approach to transfer knowledge from BERT, utilized the produced BERT embeddings (from the last hidden layers) as the input for other classification models based on ML or other deep learning methods. For example, in [25] the model used a combination of GloVe (global vectors for word representations) embeddings, BERT embeddings, and a set of psycholinguistic features to be fed into bidirectional long short-term memory (BiLSTM) Models. This is the same as in [26], where they proposed a two-stage classification architecture. At the first stage, they utilized a fine-tuned BERT to extract vector transformations after classifying ISEAR data into seven classes (i.e., anger, disgust, sadness, fear, joy, shame, and guilt). These vectors were then fed into a BiLSTM classifier for prediction. We find [18] used another approach in utilizing BERT, which is different from the above two commonly adopted approaches to transfer knowledge from BERT. They constructed an auxiliary sentence for each input and converted the task into a sentence-pair instead of a classification task. They stated that their model significantly outperformed both typical feature-based methods and fine-tuning methods and achieved new state-of-the-art performance on multi-class classification datasets. In this study, we have chosen to use SoftMax, following feedback from [27] that fine-tuning BERT for sentence classification tasks has proved to be effective in supervised sentence classification tasks.

## 3. Methodology and Approach

Our sentiment analysis approach consists of two phases (See Figure 1). The first phase consists of building our sentiment classification model, while the second phase consists of using this classification model with the NRC lexicon to detect emotions in the data. For building the model in phase one, we used a small set of training data (1766 labeled records) and a pre-trained BERT-base-uncased. The job of the model produced was to categorize any message into one of the three known sentiment classes: positive, negative, or neutral. In the second phase, we used the built model to classify a larger unlabeled dataset of 10,247 records. As a result, these 10,247 records were classified into 2597 positive, 1525 negative, and 6125 neutral. Because the objective is to detect the negative emotions, we extracted only the negatively labeled messages (1525 records) for further fine-grained emotions analysis. To fine grain the emotions, we utilized NRC lexicon to iterate through

the negative messages and count the occurrences of words associated with these four emotions: anger, disgust, fear, and sadness. The objective of counting the words per each of the four negative emotion is to identify what are the most utilized words by students to express each emotion.

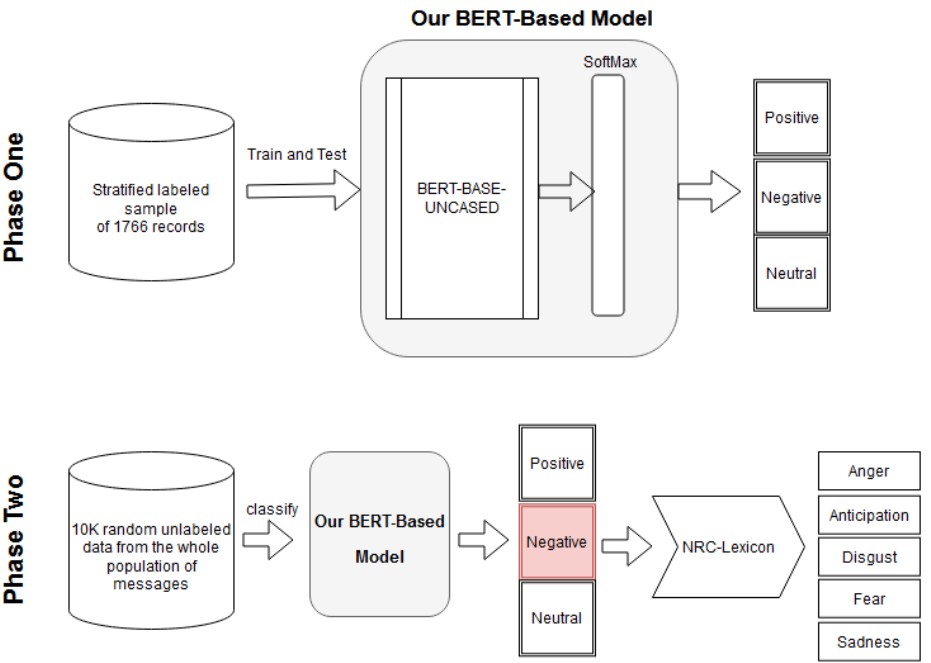

**Figure 1.** The proposed two-phase approach for emotions analysis.

*3.1. Baseline ML-Based Model*

To compare the accuracy of our BERT-based model, we built ML-based models, using the same small dataset (1766 records) for training and testing. We followed the state-of-art pre-processing and features engineering and extraction methods to build classifier utilizing different algorithms—namely SVM, Naive Bayes, logistic regression, Decision Tree, and Gaussian Bayes. The pre-processing included removing stop words, lowering the case, and replacing the emoticons and emoji with their textual representations. We used the following Python libraries to complete the tasks in this order:

1.  "Emot" and "Emoji" to convert emoticons and emojis found in the message to their textual representations. See Table 1.
2.  "Nltk" to remove non-alphabetic characters, lowercase the message, tokenize the message, and remove stop words
3.  Iterate the tokens and replace textual representations for emoticons or emoji with the associated word, i.e., a positive emoji with the word "Positive", a negative one with "Negative", and a neutral one with "Neutral". See Table 1.
4.  "Sklearn" to extract features, namely BOW, and to build the classification model using different ML algorithms.

**Table 1.** Emoji and Emoticons conversion table.

| Emoji/Emoticon | Textual Meaning | Replaced with |
| --- | --- | --- |
| ☺ | smilingface | Positive |
| ☹ | poutingface | Negative |
| :-\| | neutralface | Neutral |

We used a combinations of different features to reach the highest accuracy we can get using ML algorithms.

The state of the art models use n-gram BOW or GloVe word embeddings [25]. We started building our model using 1-gram BOW only as a single feature. The results was quite high as depicted in Table 2. Then we added other statistical features such as num of chars, or num-words had no positive contribution to the model's accuracy, therefore we decided to exclude them from the final model. Moreover, a feature we also dropped, after finding its low contribution, was the TFIDF vectorizer that we ysed to get a token matrix for the POS tags. Eventually, the features we settled on and achieved the highest accuracy was only one namely, 1-gram BOW feature.

We also built a model using Glove word embeddings. The accuracy dropped dramatically as depicted in Table 2. So, finally, we agreed on considering the model with 1-gram feature as the highest scored model to use as a benchmark when comparing to the BERT-based model.

For training the model, we used Stratified Folds with k = 5, and after reducing the number of features from 4553 to 910 (dropping 80%) using "sklearn.feature-selection", the highest accuracy we could get was 84% when using Naive Bayes.

**Table 2.** Comparative test on baseline ML algorithms using 1-gram BOW VERUS Glove word embeddings.

| | | | SVM | Naïve Bayes (NB) | Decision Tree (DT) | Logistic Regression (LR) | Gaussian NB (GNB) |
|---|---|---|---|---|---|---|---|
| Glove word embeddings | Accuracy | | 0.68 | 0.68 | 0.49 | 0.71 | 0.61 |
| | macro avg | precision | 0.33 | 0.49 | 0.33 | 0.53 | 0.53 |
| | | recall | 0.33 | 0.45 | 0.33 | 0.48 | 0.57 |
| | | f1-score | 0.32 | 0.44 | 0.32 | 0.46 | 0.5 |
| 1-gram BOW | Accuracy | | 0.75 | 0.84 | 0.65 | 0.81 | 0.75 |
| | macro avg | precision | 0.76 | 0.85 | 0.65 | 0.81 | 0.83 |
| | | recall | 0.76 | 0.83 | 0.65 | 0.80 | 0.75 |
| | | f1-score | 0.75 | 0.83 | 0.65 | 0.81 | 0.75 |

### 3.2. Data

The data represent textual messages posted by the users of an educational app in the app's public chatroom. The app brings openly licensed (CC) textbooks to high school and university students worldwide via an interactive platform. The textbooks cover different subjects such as calculus, chemistry, algebra, and biology. The users come from all around the world, and the top countries from which user come are India, Ghana, the United States, South Africa, and Nigeria. The percentage of male to female is almost equal (49%:51%). Due to the fact that the textbooks are for high school and university students, the age average for users are between 16–22 years. These characterization about the users were provided by the app owners.

Logging-in using a valid email address is mandatory for posting messages in the chatroom. The logging-in process involves using a trusted third party, hence no PII related to users are stored or processed. This condition also limit bot-generated messages. The identity or name of users is completely anonymous and is not by any mean included or hinted at in the analysis. Therefore, we had no demographic or other types of descriptive data of the users.

The chatroom, a component of the app, enables the students to interact and post questions or feedback. The messages are posted mainly in English, and our study included only those in English. The messages length vary from 1 to 60 word (or an emoji or emoticon). The average word-count is 32 words per message, and the average char-count is 135 char per message. Messages posted in the chatroom can be categorized as either objective or subjective.

The subjective category is comprised of academic questions or comments on an academic topic or subject. Such objective messages basically bear no sentiment and can be labeled as neutral. On the other hand, the subjective or sentiment-bearing are messages

expressing different kinds of emotions, e.g., anger, happiness, fear, disgust, or enthusiasm. Those can be labeled as either positive or negative. The messages can also contain emoticons, emoji, or misspelled words. Examples of such messages can be found in Figure 2.

| Class | Example of Messages |
|---|---|
| Positive | A Wonderful app 👍👍👍👍 continue |
| | I'm hoping to study abroad I'm really interested in clinical pay |
| | It's very heavy but very interesting. And it open so many ways for future. |
| Negative | Study pressure is high |
| | Did you ve depression before? |
| | Guys.. I have a problem and I'll really like to know how to overcome depression. |
| | I am afraid I'll kill myself |
| Neutral | A protein signature is? |
| | Can people die from asthma? |
| | Draw the field lines around the two negative charges |

**Figure 2.** Examples of messages.

Out of the data population, we randomly pulled two datasets. Both datasets—dataset1 and dataset2—were pulled as CSV files from the whole data population. See Figure 3 and Table 3 for details about the two datasets used. Dataset1 was the training/testing dataset, which comprised a stratified sample of 1766 annotated records and was used to train the BERT-based model. We made sure that dataset1 was stratified, and the three classes were almost equally represented as follows: negative with 611 records, positive with 616 records, and neutral with 539 records. Human annotators manually labeled Dataset1. First, two people, separately, annotated the dataset. Then the two versions were compared to check which records were annotated differently (i.e., records annotated as negative by one person while annotated as neutral by the other person). A third person then reviewed the mismatching annotated records to resolve the conflicting annotation.

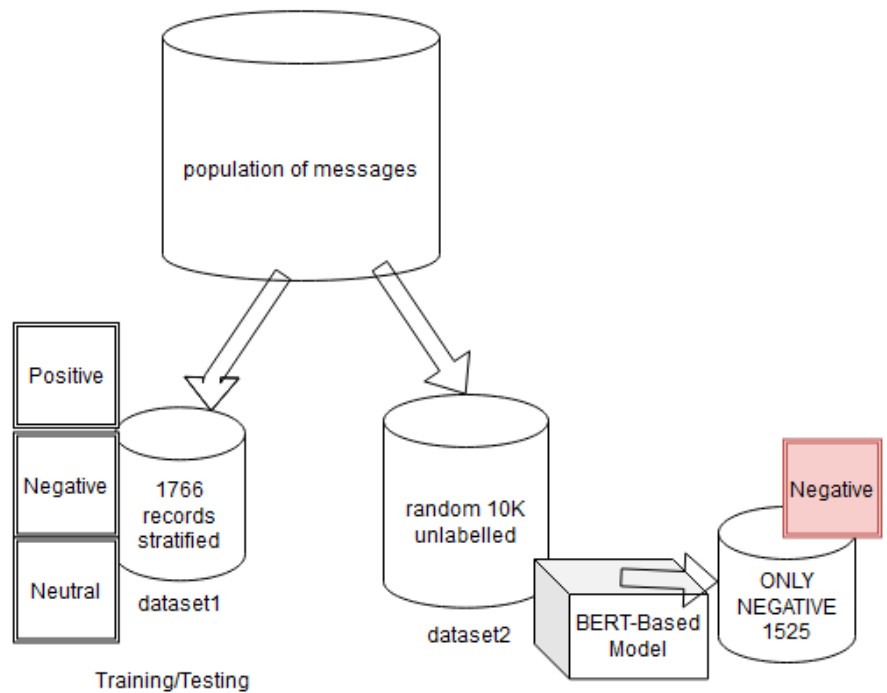

**Figure 3.** Datasets used in the approach.

The dataset2 was an unannotated larger dataset of 10,247 records. The records in dataset2 were selected randomly; we did not know the classes represented in it before

the classification. We uploaded dataset2 into the produced BERT-based model to be automatically classified. Then we sifted the results to carry on the fine-grained emotions analysis, using only the negative-classified messages. As a result, we ended up with 1529 negative-classified records.

**Table 3.** The two datasets used in the approach.

| | Used for | Size | Class Representation | | |
|---|---|---|---|---|---|
| | | | Positive | Negative | Neural |
| Dataset 1 | Training and testing the BERT-based model and the ML-based model | 1766 records | 616 | 611 | 539 |
| Dataset 2 | Automatic annotation used for fine-grained analysis | 10,247 records | 2597 | 1525 | 6125 |

*3.3. Phase One: Build BERT-Based Model*

The building blocks of a transformer are essentially a stack of encoder and decoder layers. BERT has two variants: (1) BERT Base with 12 layers (transformer blocks), 12 attention heads, and 110 million parameters, and (2) BERT Large with 24 layers (transformer blocks), 16 attention heads, and 340 million parameters. BERT was trained using 3.3 billion words in total (with 2.5B from Wikipedia and 0.8 from Books Corpus) using masked language modeling (MLM) and next sentence prediction (NSP) objectives [6]. For our model, we used BERT-base-uncased. This model does not differentiate between uppercase and lowercase English words, in contrast to BERT-base-cased. We used TFBertForSequenceClassification: which is a BERT model transformer with a sequence classification/regression head on top. For the input layer, the maximum number of tokens accepted is 512, and all inputs must have equal sequence lengths because BERT uses absolute position embeddings. Padding is used to make inputs of the same length. In our case, to avoid a long sequence of unnecessary paddings and hence wasted time in computation, we measure the average length of our data input (avg-len = 135) and truncate longer inputs to this length. The Encoder steps include:

1. Tokenize the sentence.
2. Pre-append the "[CLS]" token to the start.
3. Append the "[SEP]" token to the end.
4. Map tokens to their IDs.
5. Pad or truncate the sentence to avg-len = 135
6. Create attention masks for [PAD] tokens.

The output layer consists of a simple SoftMax classifier on the top of the BERT encoder to calculate the probability of labeling a message against pre-defined categorical labels. Fine-tuning of the target model, we use these key parameters:

- batch size: 32
- Adam Optimizer with a learning rate of $2 \times 10^{-5}$ and epsilon of $1 \times 10^{-8}$
- train epochs: 4

Working Environment: Google Colab and GPU. Source Code can be found at Github [28].

BERT-based model performance: The model converges to its highest validation accuracy of 0.87 at epoch 4 with training loss of 0.23. The testing accuracy is impressive with 0.91. In comparison with the ML-based classification model (our baseline), the BERT-based model outperforms in all performance indicators except for the "Neutral class" precision. See Table 4. for comparative performance between the two models. Note that we are comparing our BERT-based model to the highest scored ML-based model, which in our case is the Naive Bayes model.

**Table 4.** Comparative Performance Test between BERT-based model and Naive Bayes mode.

|  | Precision (%) | Recall (%) | F1-Score (%) |
|---|---|---|---|
| Negative–BERT Based | 0.92 | 0.88 | 0.90 |
| Negative–Naïve Bayes | 0.85 | 0.81 | 0.83 |
| Positive–BERT Based | 0.88 | 0.95 | 0.91 |
| Positive–Naïve Bayes | 0.77 | 0.95 | 0.58 |
| Neutral–BERT Based | 0.92 | 0.88 | 0.90 |
| Neutral–Naïve Bayes | 0.94 | 0.73 | 0.82 |

*3.4. Phase Two: Use NRC-Lexicon to Discover Fine-Grained Emotions*

After building the BERT-based model, we used it to classify a larger random unlabeled dataset of 10,247 items from the total population of messages. We then separated the negative-labeled ones for further analysis. Out of the 10,247, we found 1525 negative messages. For our further fine-grained emotion analysis, we utilized the NRC Lexicon. The NRC Emotion Lexicon is a list of 14,182 English words and their associations with eight basic emotions (anger, fear, anticipation, trust, surprise, sadness, joy, and disgust) and two sentiments (negative and positive). We iterated the 1525 negative messages to extract from each the occurrences of NRC-words associated with anger, disgust, fear, and sadness. The objective was to identify the dominant emotions and the most-utilized words to express each emotion.

## 4. Results and Discussion

The objective of our emotions detection approach is to identify the top emotions expressed by students in an educational platform and to identify which words the students have used to express these emotions. As a mean to discover such insights, we built two models, namely the ML-based model and the transfer learning-based model (BERT-based). Looking at the performance metrics in Table 4, it is obvious how the BERT-based model outperformed the ML-based model.

For our ML-based model, we had to test combinations of different features as recommended by the state-of-the-art research such as BOW and Glove word embeddings. While for the BERT-model, we didn't have to do any feature engineering. Nevertheless, the accuracy of our BERT-based model was 91% which outperformed the ML-based with accuracy of 86%. This indicates the advantage of using transfer learning to solve the shortage of "labeled" data and solve the need for features engineering that can be complex for some datasets and domains.

In terms of insights discovered, after applying our approach on the corpus of messages, we found out, as Figure 4 exhibits, that among the four negative emotions, students expressed fear more than any other emotion, followed by anger, sadness then disgust. Moreover, students expressed their emotions using a small set of vocabulary. Figures 5–8, show the top words utilized by the students to express fear, anger, sadness, and disgust respectively. For example, in both fear and sadness, students used a set of limited to 16 words. Words "suicide, anxiety, bad, confusion" were among the top used words across the different negative emotions. This common occurrences for the same words in more than one emotions (e.g., word suicide is associated with fear and also with anger) is due to the NRC categorization and association of words to emotions. Focusing on the small set of keywords can remarkably enhance the detection in other content analyses of such emotions and, hence, can improve the educator-designed supportive interventions. Educators should not overlook such expressive words when these are raised by their students. Educators can proactively track the usage of these words in each class as a metric to evaluate the

instructor, the way of teaching, and the load of work. Instead of waiting till the end of the semester for students' feedback, a proactive approach can be adopted along the semester.

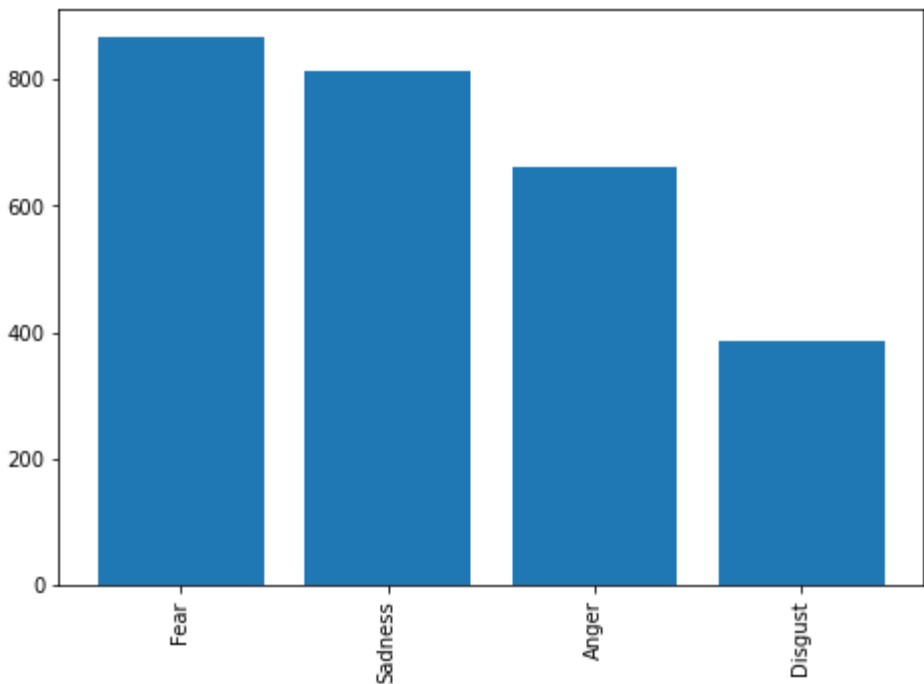

**Figure 4.** The order of the dominant negative emotions detected.

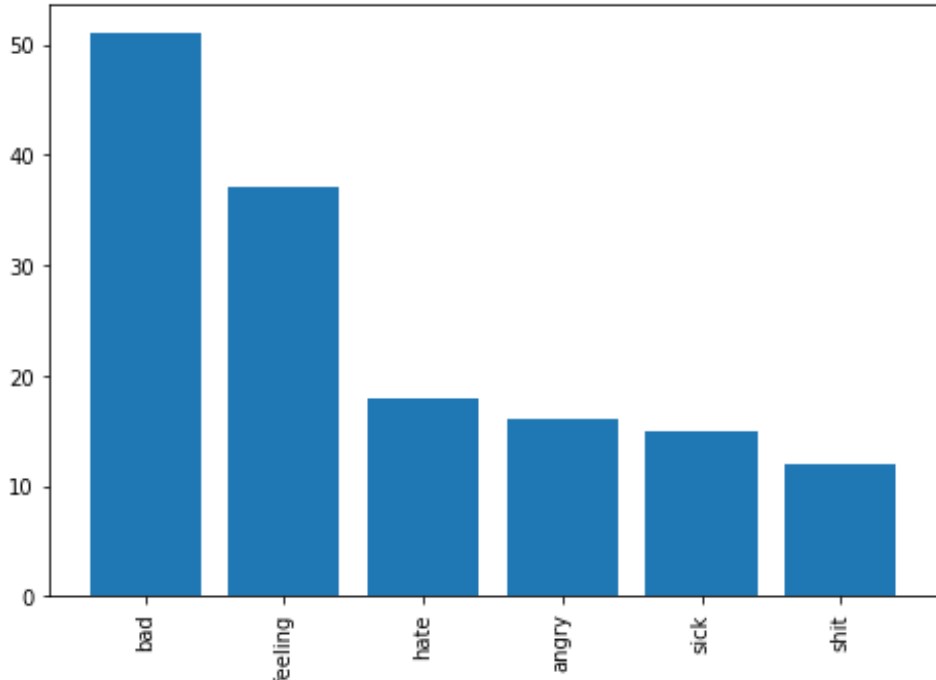

**Figure 5.** The most utilized words expressing disgust.

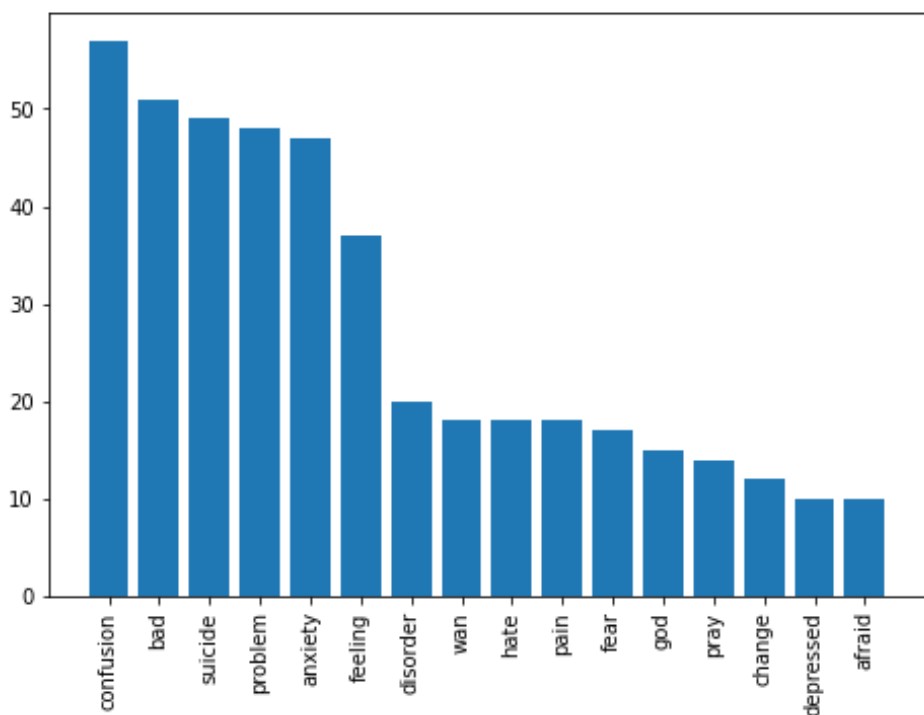

**Figure 6.** The most utilized words expressing fear.

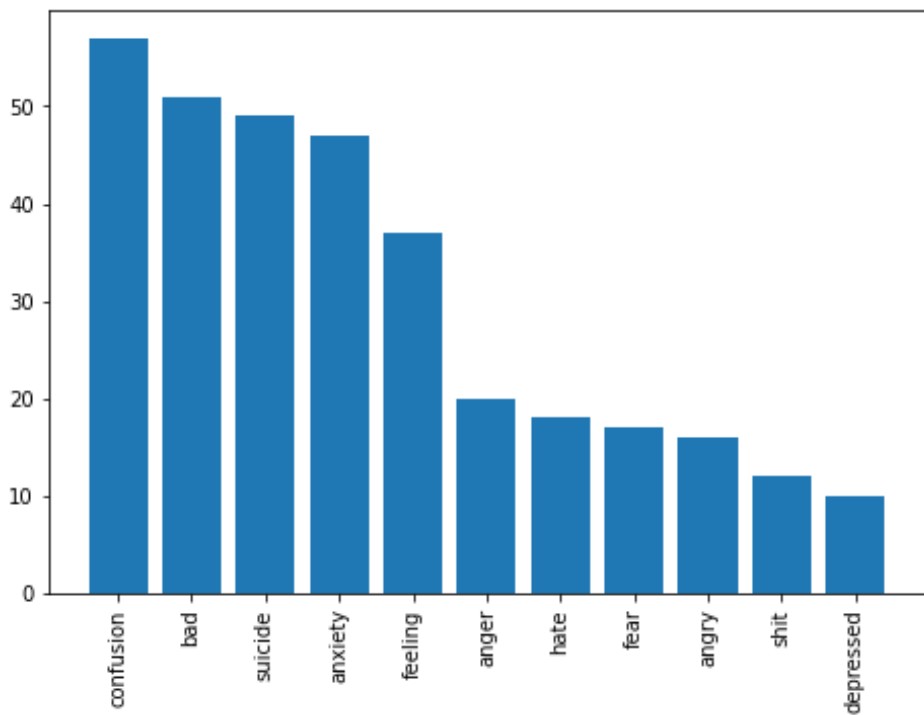

**Figure 7.** The most utilized words expressing anger.

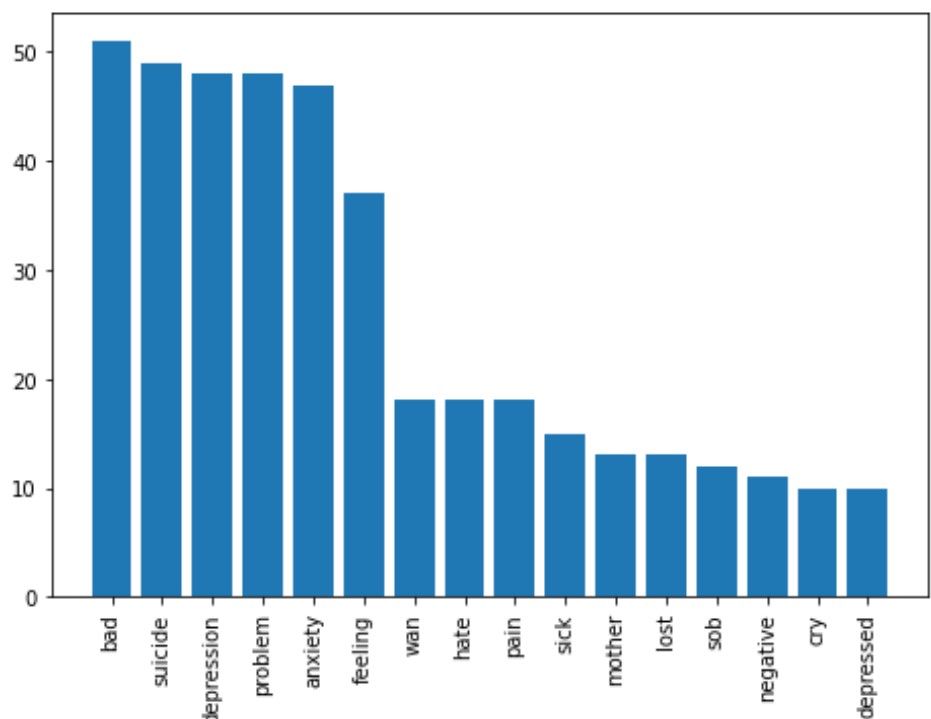

**Figure 8.** The most utilized words expressing sadness.

## 5. Conclusions and Future Work

It is evident that transfer learning using a pre-trained BERT model significantly increased prediction accuracy with only a few training epochs and a small dataset. We trained the model to do sentiment classification, which is only half the job of detecting fine-grained emotions. We used in tandem a lexicon to assist in the process and break down the negative sentiments into precise emotions. Nevertheless, the potential is high for transfer learning to fully automate and assign the task to one model instead of breaking it down as we did. We hope to continue exploring the power of BERT to develop innovative approaches to further enhance our system. Moreover, we plan to investigate the implementation of transfer-learning and federated learning (FL) in education and eHealth systems, where privacy is highly enforced. Analysis of emotions in domains such as education and health can greatly contribute to the overall strategy of improving related services. The long term goal is to evaluate whether this classification can determine student mental health status. Nevertheless, the privacy-preserving conditions in such domains require innovative approaches. At this point in time, few researchers have utilized FL in emotions analysis. The work of [29], for example, relied on FL and IoT applications for analyzing emotions in an office environment. Similarly, we envision utilization of emotions analysis in autonomous driving, where sensing riders' feelings can provide feedback to enhance the journey. Our study focused on textual inputs; an expanded work can include written, facial, or audible expressions. Moreover, we plan to compare BERT with other transformer like GPT-3 and XLNet.

**Author Contributions:** Conceptualization, N.A., N.Z. and H.I.; methodology, N.A., N.Z. and H.I.; software, N.A.; validation, N.A., N.Z. and H.I.; Data collection and cleaning, N.A.; writing original draft preparation, N.A.; writing review and editing, N.A., N.Z., H.I. and M.K.; visualization, N.A.; supervision, N.Z. and M.K.; All authors have read and agreed to the published version of the manuscript.

**Funding:** This research received no external funding.

**Institutional Review Board Statement:** Not applicable.

**Informed Consent Statement:** Not applicable.

**Data Availability Statement:** Data can be shared upon request to the corresponding author.

**Conflicts of Interest:** The authors declare no conflict of interest.

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
