# Peer review of "Detecting Emotions behind the Screen"

_ai, doi:10.3390/ai3040056_

Round 1

Reviewer 1 Report

Thanks for submitting your work to the journal. It was easy to read and understand, and I appreciate spending time on understanding students' reactions in a digital world like MOOC. The authors propose a BERT-based model (transfer learning) to determine student emotions on the sampled dataset they created from a MOOC course. The authors then evaluate the model on a much larger dataset to see the efficacy of the model in detecting student emotions, particularly negative emotions. 

Questions to answer: 

1. A three-level classification task for the BERT model is trivial. It is unclear why not do 7-level (all negative emotions as separate classes) classification directly without using the NRC lexicon model. How does the accuracy change? You might have to train for more epochs, but the benefit of NRC on top of BERT is very unclear. 

2. The second biggest challenge for me to understand is - why not use an MLP model? Ofcourse MLP requires a feature embedding vector, which you can generate using a naive BERT or GloVe. 

3. In the evaluation, why does BERT not perform as well as the Naive Bayes design for a neutral setup? What is the intuition for its lower performance? 

4. Finally, future work can also consider looking at GNN-based design, which might do a better job. 

5. Will you be open-sourcing the input dataset used for the training and testing? 

Author Response

Reviewer 1: Comments and Suggestions for Authors

Thanks for submitting your work to the journal. It was easy to read and understand, and I appreciate spending time on understanding students' reactions in a digital world like MOOC. The authors propose a BERT-based model (transfer learning) to determine student emotions on the sampled dataset they created from a MOOC course. The authors then evaluate the model on a much larger dataset to see the efficacy of the model in detecting student emotions, particularly negative emotions. 

 Questions to answer: 

  1. A three-level classification task for the BERT model is trivial. It is unclear why not do 7-level (all negative emotions as separate classes) classification directly without using the NRC lexicon model. How does the accuracy change? You might have to train for more epochs, but the benefit of NRC on top of BERT is very unclear. 

We appreciate the reviewer’s valuable feedback and interest in understanding more about our approach. We are aiming with this work to help identify the minimal vocabularies used by today’s online students that would express their negative emotions in high school/college online platforms. We are introducing another angle to detect emotions but identifying the most used words among a specific young population between 16 - 22 years. We opted to use NRC to leverage its database of words used for four negative emotions (fear, anger, disgust, and sadness). However, before this refinement we relied on BERT first to give me the highest accurate classification for a negative sentiment. NRC then followed to come up with dictionary used among this population.  However, we find the reviewer’s point valid in eliminating the NRC and explore BERT’s capabilities in building up this dictionary, which is something we will consider in our future work. Such approach would required a labeled dataset with 7 different classes which we didn’t have available for this study.

  1. The second biggest challenge for me to understand is - why not use an MLP model? Ofcourse MLP requires a feature embedding vector, which you can generate using a naive BERT or GloVe. 

We thank the reviewer for the valuable feedback and we believe it is important to include it in our future work.

  1. In the evaluation, why does BERT not perform as well as the Naive Bayes design for a neutral setup? What is the intuition for its lower performance? 

This is a very valuable observation. We believe that this is because, for BERT training, we had to truncate longer inputs to the average length (avg-len=135) (more in lines 236  - 240) which might have caused some key words to be dropped. Such words might have balanced the neutrality of the sentence.

  1. Finally, future work can also consider looking at GNN-based design, which might do a better job. 

Thank you for this valuable suggestion. We will consider this in our future work

  1. Will you be open-sourcing the input dataset used for the training and testing? 

We are still working with the platform owner to publish the dataset.  However, if you are interested in getting and working on the datasets, please contact the platform owner at play.google@quizover.com

Their online apps are found here: https://play.google.com/store/apps/developer?id=QuizOver.com

If you download any app, you fill find these messages of students published in the chatbox.

Reviewer 2 Report

The topic of study of emotions and their relationship with academic performance seems very interesting to me

Even though it is well understood. Perhaps the summary should explain the context a little better. The summary must clearly include the objective, the method (participants, context), procedure, data collection instruments, analyzes performed and results.

The phases carried out in the methodology section are of great interest

the accuracy of our BERT-based model as well as the ML-based models, I think is timely, as is the use of different algorithms in the study such as SVM, Naive Bayes, logistic regression, decision tree

the accuracy of our BERT-based model as well as the ML-based models, I think is timely, as is the use of different algorithms in the study such as SVM, Naive Bayes, logistic regression, decision tree

Data and calculations are fine.

The section for presentation of results and discussions is fine

It is an excellent work that should be published both for the subject of study and for the procedure followed

Author Response

Reviewer 2: Comments and Suggestions for Authors

The topic of study of emotions and their relationship with academic performance seems very interesting to me. Even though it is well understood. Perhaps the summary should explain the context a little better. The summary must clearly include the objective, the method (participants, context), procedure, data collection instruments, analyzes performed and results.

The phases carried out in the methodology section are of great interest

the accuracy of our BERT-based model as well as the ML-based models, I think is timely, as is the use of different algorithms in the study such as SVM, Naive Bayes, logistic regression, decision tree

Data and calculations are fine.

The section for presentation of results and discussions is fine

It is an excellent work that should be published both for the subject of study and for the procedure followed

We highly appreciate the reviewer’s time in reviewing our manuscript and for the reviewer’s valuable feedback. We find the supportive feedback highly encouraging for us to continue enhancing our work on this topic. Thank you so much.

Reviewer 3 Report

This manuscript presents a two-step framework for emotional detection. It uses fine tuned BERT model for sentiment classification and uses the NRC emotion lexicon for fine-grained detection. 

1. As the authors claimed, there are a few publications based on MOOCs, and their work is the first study based on online educational platforms. However, the differences between online educational platforms and MOOCs is not very clear to me. What makes emotional detection on online educational platforms distinguished to on MOOCs?

2. In related works, the authors discussed a few studies adopting BERT, which is similar to what the proposed framework does. As being said, the innovation and novelty of this work is not very clear to me.

3. The section “Baseline ML-based Model” was not written clearly.

a) In line 165-172, how many features are finally used to train the classic ML methods? 

b) In line 173, neither table 1 or figure 1 has the performance to support the statement “The accuracy dropped dramatically as depicted in Table 1”.

4. Although the data platform mentions they provide nearly 1:1 gender ratio records, how is the ratio in the sampled training set (dataset1 and dataset2)? There is very limited information provided for the actual training data in this paper.

5. The training epochs mentioned in line 252 and line 257 are different.

6. Formatting must be improved. 

a) Figure 5 is too small, making it hard to read.

b) Some figures are tables (figure 6, 7)

c) Figures should be placed before the References section.

Author Response

Reviewer 3: Comments and Suggestions for Authors

This manuscript presents a two-step framework for emotional detection. It uses fine-tuned BERT model for sentiment classification and uses the NRC emotion lexicon for fine-grained detection. 

  1. As the authors claimed, there are a few publications based on MOOCs, and their work is the first study based on online educational platforms. However, the differences between online educational platforms and MOOCs is not very clear to me. What makes emotional detection on online educational platforms distinguished to on MOOCs?

First, we want to thank you for your kind review to our manuscript and the valuable feedback and suggestions.

Regarding the difference between a MOOC platform and the online education platform which we used is that our platform has more resemblance to a college or a university online platform in which a group of students would share a one semester journey together attending a specific class. This shared time-window would develop a more of a community among students. This community has an impact on students’ openness about expressing their emotions, frustrations, or pain.

On the other hand, the online MOOCs, are in principle are open for new subscribers all year 24 hours. There is no drive here for a community building among users, at least not to share emotions.

Therefore, we believe that the data from our online platform is closer to high schoolers/college online platforms. In the papers we reviewed, we couldn’t find similar approach in terms of data source (“NONE MOOC” platforms) and in objective (as elaborated below). We hope we could explain why we think the data from such platform has uniqueness not found in MOOC platforms.

  1. In related works, the authors discussed a few studies adopting BERT, which is similar to what the proposed framework does. As being said, the innovation and novelty of this work is not very clear to me.

We appreciate the reviewer’s interest in understanding more about our work and its contribution. We are aiming with this work to help identify the minimal vocabularies used by today’s online students that would express their emotions in high school/college online platforms. We are introducing another angle to detect emotions but identifying the most used words among a specific young population between 16 - 22 years. Our work can be further enhanced in the future work by comparing BERT to other transformers accuracy or by shortening the pipeline by eliminating the NRC usage.

  1. The section “Baseline ML-based Model” was not written clearly.
  2. a) In line 165-172, how many features are finally used to train the classic ML methods? 

Thank you for highlighting this missing information. We’ve added it in lines 172-173.

Basically, we tried different features but settled with one only namely, the 1-gram BOW.

  1. b) In line 173, neither table 1 or figure 1 has the performance to support the statement “The accuracy dropped dramatically as depicted in Table 1”.

Thank you for highlighting this mistake. We fixed it by referring to the right Table (which is Table 2)

  1. Although the data platform mentions they provide nearly 1:1 gender ratio records, how is the ratio in the sampled training set (dataset1 and dataset2)? There is very limited information provided for the actual training data in this paper.

Thank you so much for this on-spot observation. Although this feature (gender) was not considered in the sampling process, we find it important to reconsider it in our future work.

  1. The training epochs mentioned in line 252 and line 257 are different.

Thank you for spotting this inconsistency. We have fixed it in line 257 to be epoch 4.

  1. Formatting must be improved. 
  2. a) Figure 5 is too small, making it hard to read.

We agree with the reviewer on this suggestion to enhance the clarity. Therefore, we enlarged the figure to be more readable.

  1. b) Some figures are tables (figure 6, 7)

We agree with the reviewer on this suggestion to enhance the resolution of the tables. Therefore, we changed these figures into Tables.

  1. c) Figures should be placed before the References section.

This has been fixed. All Tables and Figures appear now before the Reference Section.

Thank you for spotting this.

Reviewer 4 Report

Alkaabi et al. used BERT natural language analyzing to predict the emotional health of students. Overall, this transformer seems to work well. The topic is also very important in this digital era with a variety of social media apps. However, the comparison between BERT and basic ML methods is redundant. It is well-known that transformers like BERT are way better than those classical ML algorisms in NPL. Even though, this is still a good writing manuscript for this journal. I only have several points:

1. Fig. 1 and 2 are not necessary. You can combine them with Fig. 7 to show BERT are better than all the classical ML algorisms.

2. It makes more sense to compare BERT with other transformer like GPT-3 and XLNet.

3. How the length of text affects the accuracy of you model?

4. For NPL, the training dataset is too small.

Author Response

Reviewer 4: Comments and Suggestions for Authors

Alkaabi et al. used BERT natural language analyzing to predict the emotional health of students. Overall, this transformer seems to work well. The topic is also very important in this digital era with a variety of social media apps. However, the comparison between BERT and basic ML methods is redundant. It is well-known that transformers like BERT are way better than those classical ML algorisms in NPL. Even though, this is still a good writing manuscript for this journal. I only have several points:

  1. 1 and 2 are not necessary. You can combine them with Fig. 7 to show BERT are better than all the classical ML algorisms.

We highly appreciate your kind review to our manuscript and for your valuable feedback. We took this advice in consideration and combined the two figures below (1,2) and transformed it into a Table instead a figure to enhance the readability. Now it is Table 2.

  1. It makes more sense to compare BERT with other transformer like GPT-3 and XLNet.
  2. How the length of text affects the accuracy of you model?
  3. For NPL, the training dataset is too small.

For points #2,3 and 4: we thank the reviewer for these valuable suggestions and we find them important to consider in our Future work. We reflected this in Section. 4. Conclusion and Future Work

Round 2

Reviewer 3 Report

The authors addressed most of my concerns and the readability of section 2 is improved a lot.

Line 275 - 400 are not presented in English. 

Author Response

We profoundly thank the reviewer for the time spent on reviewing our manuscript and we want to apologize for this mistake concering the "non-English" text embedded in the work. We have removed it and it is not represented any more in the file.

We have also did a second review for the English, but we are more than happy to put it under a review by a native-English speaker for further enhancement if recommended by the reviewer.

Round 3

Reviewer 3 Report

I received the manuscript titled with v3, but the non-English text issue from line 275 to 400 has not been changed. Once the authors fix this issue, I believe it is acceptable for publishing.

Author Response

Dear Reviewer,

our apologies for this mistake.

We have reattached the correct version in PDF.

Lines with non-english words are not included now and lines line 275 to 400 reads well now.

thank you,
